# Laser Pulses for Studying Photoactive Spin Centers with EPR

**DOI:** 10.3390/mi16040396

**Published:** 2025-03-28

**Authors:** George Mamin, Ekaterina Dmitrieva, Fadis Murzakhanov, Margarita Sadovnikova, Sergey Nagalyuk, Marat Gafurov

**Affiliations:** 1Institute of Physics, Kazan Federal University, Kremlevskaya 18, 420008 Kazan, Russia; georgemamin@gmail.com (G.M.); dev600@mail.ru (E.D.); margaritaasadov@gmail.com (M.S.); marat.gafurov@kpfu.ru (M.G.); 2Ioffe Institute, Russian Academy of Sciences, Polytechnicheskaya 26, 194021 St. Petersburg, Russia; snagalyuk@gmail.com

**Keywords:** microcontroller, optical spin polarization, electron paramagnetic resonance

## Abstract

Quantum technologies are currently being explored for various applications, including computing, secure communication, and sensor technology. A critical aspect of achieving high-fidelity spin manipulations in quantum devices is the controlled optical initialization of electron spins. This paper introduces a low-cost programming scheme based on a 32-bit STM32F373 microcontroller, aimed at facilitating high-precision measurements of optically active solid-state spin centers within semiconductor crystals (SiC, hBN, and diamond) utilizing a multi-pulse sequence. The effective shaping of short optical pulses across semiconductor and solid-state lasers, covering the visible to near-infrared range (405–1064 nm), has been validated through photoinduced electron paramagnetic resonance (EPR) and electron nuclear double resonance (ENDOR) spectroscopies. The application of pulsed laser irradiation influences the EPR relaxation parameters associated with spin centers, which are crucial for advancements in quantum computing. The presented experimental approach facilitates the investigation of weak electron–nuclear interactions in crystals, a key factor in the development of quantum memory utilizing nuclear qubits.

## 1. Introduction

Solid-state spin defects are garnering significant interest within the scientific community as potential qubits, placing them in competition with other technologies such as Josephson junctions (superconducting qubits), ultra-cold atoms, trapped ions, and polarized photons [1]. The generation of defects in semiconductors through techniques like high-energy electron [2] or proton irradiation [3], ion implantation [4], and femtosecond laser pulses [5] demonstrates the relative ease of their production. The ability to execute quantum manipulations with spin defects at room temperature not only simplifies the implementation of quantum algorithms but also facilitates the maintenance of the technical platform [6]. Wide-band-gap semiconductors, such as diamond, silicon carbide (SiC), and hexagonal boron nitride (hBN), provide a broad range of point defects and energy levels. These defects (color centers) significantly alter the optical properties of the host material where spin states can be reliably initialized through laser exposure. The remarkable spin–optical and coherent properties of defects have led researchers to explore their potential as qubits in the realm of quantum information processing [7,8,9,10,11,12,13,14,15,16].

Vacancy defects in SiC (e.g., silicon vacancies and divacancies) exhibit high spin states (with electronic spins of *S* = 1 and *S* = 3/2) and long coherence times, enabling quantum applications [8,9]. Their infrared luminescence aligns with fiber-optic (telecommunications) and biosensing windows [3,6]. Similarly, 2D hexagonal boron nitride (hBN), a graphene-like direct semiconductor, hosts spin-active defects like boron vacancies (like $V_{B}^{-}$, *S* = 1, excitation wavelength, λ_exc_ = 532 nm), which is promising for quantum interfaces and sensors due to environmental sensitivity [7,16]. Both materials undergo photoactivation via intersystem crossing (ground ^3^A_2_ → excited ^3^E states), with nonradiative recombination through a metastable ^1^A state and preferential population of the ground state with *M*_S_ = 0, (*M*_S_ is the quantum number of the electron spin projection).

One suitable method for studying color centers is electron paramagnetic resonance (EPR), which induces microwave transitions between different magnetic sublevels. The EPR approach has a high sensitivity, with the ability to detect signals from impurity paramagnetic centers with low concentrations (10^9^ spins/G). Conducting experiments in pulsed mode using various optical and microwave pulse sequences makes it possible to study the dynamic characteristics of spin defects such as spin–spin/dephasing/decoherence (*T*_2_) and spin-lattice (*T*_1_) relaxation times [17]. In the case of photoinduced spin centers, a simultaneous supply of optical excitation must be provided, together with microwave radiation, to the electron spin initialization. The extensive range of potential charged defects in studied crystals complicates the identification of qubits through experimental means. The presence of ionized and de-ionized defects creates a fluctuating electrical environment that affects the optical fine structure of quantum emitters, leading to unpredictable shifts in the wavelength of the zero-phonon lines (ZPLs) and contributing to inhomogeneous line broadening, commonly known as spectral diffusion [18]. Continuous laser radiation has several negative effects on the coherent properties of spin qubits, reducing *T*_2_, which is vital for their performance in quantum computing and information processing. Background charge noise fluctuations can lead to significant gate errors and/or decoherence in semiconductor-based electron spin qubits through inter-qubit exchange coupling, which limits the scalability of exchange-based spin quantum computation. Charge noise represents a crucial decoherence channel for exchange-coupled spin qubits, necessitating further advances in device design and fabrication to reduce the sensitivity of exchange coupling to charge fluctuations [19,20,21]. Non-stable and uniform laser intensity can result in variations in the Rabi oscillation frequency, which affects control over qubit rotations [22,23]. Overall, the abovementioned points can lead to errors in gate operations and reduced fidelity. Therefore, the technical conditions for the selective photoactivation of a spin defect must be carefully chosen to avoid affecting nearby centers with variable charge states.

To mitigate these exposure effects on studied spin systems, careful control of laser parameters, such as frequency, intensity, and pulse duration, is essential, along with employing techniques like dynamical decoupling and error correction protocols in quantum computing applications. In this work, an easy-to-use pulse sequence programmer was developed using cheap and affordable components that convert continuous optical irradiation into a pulsed radiation supply mode synchronized with a sequence of microwave and radio frequency pulses. The assembled electronic circuit can be integrated with both semiconductor and solid-state lasers across a wide range of wavelengths (405–1064 nm), enabling the study of the dynamics of optically addressable spin centers through photoinduced EPR spectroscopy. The results were tested on three different solid-state matrices (SiC, diamond, and hBN) containing color centers (*S* = 1) using an additional electron–nuclear double resonance (ENDOR) method.

## 2. Materials and Methods

### 2.1. Methodology

The magnetic resonance experiments were carried out with a Bruker Elexsys E680 commercial spectrometer (Bruker, Karlsruhe, Germany) operated at a microwave frequency of 94 GHz (W-band) in a pulsed mode. The EPR spectra were acquired by detecting the amplitude of the primary electron spin echo (ESE) as a function of the magnetic field sweep, *B*, using a pulse sequence, π/2 − τ − π − τ − *ESE*, where π/2 = 40 ns and τ = 240 ns. The microwave pulse duration was chosen to be as short as possible, 36–44 ns, so the pulse excitation spectrum was about 0.9 mT. The relaxation times, *T*_2_, were measured with the Hahn sequence given above, where τ changes with a minimal step of 4 ns. Short nanosecond-scale microwave pulses required a 1 kW amplifier to achieve a 90- or 180-degree spin magnetization turn in a rotational coordinate system.

For the main experimental EPR section of this article, lasers with wavelengths of 980 nm (diode laser) and 532 nm (diode-pumped solid-state) were employed, both of which were manufactured by Changchun New Industries (CNI) Optoelectronics Tech. Co., Ltd. (China). Fiber-optic cables were used to deliver optical excitation from lasers to the resonator system. Pulse programmer descriptions are provided below in Section 3.1, “Implementation of a pulse sequence programmer”, and Section 3.2, “Formation of laser diode pulses”, in detail.

ENDOR spectra were obtained utilizing the Mims pulse sequence (π_MW_/2 − τ − π_MW_/2 − π_RF_ − π_MW_/2 − τ − *ESE*) with a 150 W radio frequency (RF) generator, where π_MW_ = 72 ns and π_RF_ = 18 µs. A satisfactory signal-to-noise ratio was ensured through the multi-scan recording (3072 scans) of the ENDOR spectrum within a reasonable period (30 min–2 h). In all experiments in the pulsed mode, the width of the rectangular pulse in the frequency range was 12.5 MHz (or 4.5 G), which was sufficient for selective resonant excitation.

In order to provide isolated spin sublevels due to the Zeeman effect, a superconducting magnet capable of creating a magnetic induction of up to *B* = 6 T was used. Low-temperature experiments were carried out using a flow helium cryostat and temperature controllers to ensure the stable measurement of EPR spectra and relaxation curves (Figure 1 (left)).

### 2.2. Samples

Semiconductor crystal 6H-SiC, with isotopic enrichment in ^28^Si nuclei (nuclear spin *I* = 0), was grown by high-temperature sublimation technology from the gas phase (physical vapor transport, PVT). 6H-^28^SiC samples were irradiated with electrons (2 MeV, 4 × 10^18^ cm^−2^) and annealed at *T* = 900 °C in an argon atmosphere for 2 h. The sample size was 0.45 mm × 0.45 mm × 0.67 mm^3^ (see [15,17,24,25,26,27] for details).

A sample of 2 × 1 × 0.30 mm^3^ diamond single crystal was fabricated commercially by Element Six using HPHT (high-pressure high-temperature) synthesis. The initial concentration of the nitrogen impurities in the sample was estimated to be 5 × 10^18^ cm^−3^. The crystal was subjected to electron irradiation (2 MeV) with a flux density of 10^18^ cm^−2^, followed by annealing in a hydrogen atmosphere at *T* = 800 °C for 2 h.

hBN single crystals with dimensions of 900 μm × 540 μm × 55 μm were used in this study, commercially produced by the HQ Graphene Company (HQ Graphene, Groningen, The Netherlands) (Figure 1 (right)). The samples were irradiated at room temperature with 2 MeV electrons to a total dose of 6 × 10^18^ cm^−2^. No annealing treatments were applied to the irradiated samples.

## 3. Results and Discussion

### 3.1. Implementation of a Pulse Sequence Programmer

The implementation of a programmer implied the use of a minimum number of cheap and widespread affordable elements; therefore, the STM32F373 microcontroller (ST Microelectronics, Geneva, Switzerland) from STMicroelectronics was used as the programmer core, and a general diagram of the device is provided in [28]. The STM32F373 microcontroller operates by executing instructions from its ARM Cortex-M3 core (Arm Developer, Cambridge, UK), which runs at speeds of up to 72 MHz, utilizing a mix of 16-bit and 32-bit Thumb-2 instructions for efficient performance and compact code size. Upon power-up or reset, the system initializes its clock configuration, memory management, and peripheral settings, transitioning into the main application loop, where it continuously monitors for events such as interrupts, timer expirations, or external signals. When an event occurs, the microcontroller temporarily halts the main loop to execute the corresponding interrupt service routine, allowing for the responsive handling of tasks like data acquisition or communication. The architecture supports various low-power modes to optimize energy consumption during inactivity, while integrated peripherals—including GPIOs, timers, communication interfaces (I2C, SPI, and USART), and ADCs—facilitate interaction with external devices. The development process is streamlined through STM32Cube libraries and IDEs, enabling developers to efficiently configure hardware and implement application logic while leveraging debugging features for real-time monitoring and troubleshooting.

The scheme employs an encoder to facilitate the manual entry of parameters, utilizing a touch button designated as “Parameters” to switch into the parameter input mode, complemented by a 0.96-inch OLED screen. Engaging the Parameters button and rotating the encoder permits the selection of an editable parameter from the options listed in Table 1, which can then be altered by adjusting the encoder knob. Furthermore, pressing and rotating the encoder simultaneously allows for the modification of the step size for the current parameter adjustment. Parameters may also be configured through a USB interface that implements a virtual COM port. To establish parameters, a command line is dispatched that includes the keywords in Table 1, the “=” symbol, the parameter value, and a line feed character. For parameter retrieval, a similar command line is utilized, with the parameter value replaced by a “?” symbol. The functioning of the port is signaled by a yellow LED. The programmer case was printed with a 3D printer, as shown in Figure 2. The programmer parameters are shown in Table 1. The “Increment” column suggests the minimum step size with which the corresponding parameter of the pulse or the entire sequence specified in the rows can change. Owing to the use of the same timer to generate the position and pulse duration, the minimum pulse duration can be increased with a long pulse start time.

Complete schematic diagrams of the pulse sequence programmer used to perform the multi-pulse experiments are shown in Figure 3 (upper panel). In this system, four microcontroller timers are utilized to generate stable pulses, independent of the CPU load. The first three timers are 32-bit, while the fourth is 16-bit. The primary timer initiates the other timers responsible for pulse generation. Additionally, it features two comparators that produce a synchronizing pulse to commence the pulse sequence. The second timer generates the first pulse, the third timer produces the second pulse, and the fourth timer creates the third pulse. To facilitate the generation of three pulses along with a synchronizing pulse, a circuit containing four x-OR gates is employed. The signals from the timer comparators create the rising and falling edges of the pulses (as illustrated in Figure 3 (bottom panel)). These pulses are then combined using elements from a dual OR gate circuit. Furthermore, a switchable power supply allows for the generation of pulses with amplitudes of either 3.3 V or 5 V, depending on the configuration of the microcircuits.

The third-element 2 OR is used as a buffer to implement laser power control using pulse-width modulation (PWM). PWM is a technique used to encode information in the form of variable-width pulses, allowing for control over the power supplied to electrical devices. By varying the duration of the “on” time relative to the “off” time within a fixed period, PWM can effectively regulate the average voltage and current delivered to a load, such as motors, LEDs, or heating elements. This method is widely utilized in applications ranging from motor speed control and light dimming to signal generation and audio synthesis, offering an efficient means of managing power without significant energy loss. The flexibility and precision of PWM make it a valuable tool in embedded systems and digital signal processing. The additional fourth-element 2 OR works only as a buffer for the external synchronization circuit for starting the sequence, which turns on several programmers in a cascade.

### 3.2. Formation of Laser Diode Pulses

The programmer was linked to the pulse formation input of multiple laser sources, as detailed in Table 2. The laser beam was aimed at a BPW20RF photodiode (Vishay Intertechnology, Malvern, PA, USA) operating in a photovoltaic mode (voltage generation), exhibiting characteristic rise and fall times of approximately 4 μs [29]. The BPW20RF photodiode operates based on the principle of converting light into an electrical current through the photoelectric effect. When photons strike the semiconductor material of the photodiode, they generate electron–hole pairs, leading to a flow of current proportional to the intensity of the incident light. The device is designed to be sensitive to a wide range of wavelengths, particularly in the infrared spectrum, making it suitable for applications like remote control systems and light detection. The output current can be amplified using external circuitry to enhance sensitivity and facilitate integration with microcontrollers or other processing units. The BPW20RF features a built-in lens that focuses incoming light onto the active area, thereby increasing its effective sensitivity and ensuring efficient light collection. Additionally, its compact size and low power consumption make it ideal for battery-operated devices. The operational algorithm typically involves capturing the generated current, converting it into a voltage signal through a transimpedance amplifier, and then processing this signal to determine the intensity of the detected light, allowing for precise control and monitoring in various applications. Current variations were monitored using a Tektronix MSO 3034 digital oscilloscope (Tektronix, Beaverton, OR, USA). The duration of the programmer pulse was determined by the time required to achieve maximum power. This approach enabled the documentation of the temporal parameters for the laser units presented in Table 2. A small part of the main program code used to control the programmer is shown in Figure 4.

As can be seen from Table 2, most lasers generate pulses with a delay of about 100 μs, which must be taken into account when starting an EPR spectrometer microwave pulse sequence. By shifting the start of the programmer sync pulse to the end of the laser pulse (as shown in Figure 5), it is possible to ensure that measurements will be performed in the absence of light exposure but as close as possible to the laser pulse boundary. Also, for *NV* defects in diamond, hBN, and SiC crystals [30], the longitudinal relaxation time is significantly greater than 100 μs, so the loss of polarization due to the non-ideality of the laser pulse fronts can be considered insignificant.

### 3.3. Experimental Applications

*NV* optically polarized spin centers in the 6H-SiC crystal were chosen as an object for measuring the effect of laser radiation on the spin system coherence. The EPR properties of these systems under continuous laser irradiation are described in detail in refs. [25,27]. The upper inset of Figure 6 shows that the integrated intensity of the EPR signal (normalized value) strongly depends on the level of intensity of the incident light. It can be assumed that optical excitation can have a significant effect on the dynamic characteristics of spin defects. To estimate the heating of the sample under the light influence, we neglect the change in heat capacity with temperature and use the density, *ρ* = 3.21 g/cm^3^, and heat capacity, *C* = 690 J/(kg × K), defined in [31]. Thus, at *P* = 500 mW, the heating rate of the sample will be about 0.3 K/ms. In the case of the pulse mode with the maximum pulse duration used in the work (9 ms), the heating of the sample will be about 3 K, which can be neglected with a pulse duty cycle of less than 5%. In the continuous laser radiation mode, at *P*_out_ = 500 mW, it was found that the crystal under study is locally heated by 40 K, which has a strong critical effect on the resonance detection conditions and relaxation times due to phonon oscillation modes. In this regard, the power of the continuous laser radiation incident on the sample was reduced to 50 mW, and the heating of the thermocouple junction, in this case, did not exceed 4 K. The choice of such modes of laser action on the sample made it possible to neglect the change in relaxation times due to its heating.

The effect of laser radiation on the phase coherence time of the spin packet was estimated, which is closely related to the quantum coherence time of the spin system. For this purpose, the dependences of the transverse magnetization decay on the delay time between pulses were measured for two temperatures of 50 K and 175 K using a two-pulse Hahn sequence, as shown in Figure 6. The black squares in Figure 6 show the behavior of the ESE integral amplitude in the case of continuous laser operation, and the red circles show the dependence in the case of measurements after a laser pulse of 2 ms duration for 50 K and 200 ms for 175 K. The sequence was started with a DAF of 30 μs after the trailing edge of the laser pulse was recorded by a BPW20RF photodiode (Vishay Intertechnology, Malvern, PA, USA). As can be seen from Figure 6, the time dependences are well approximated by a single exponential process, and the phase coherence times of the spin packet found from the approximation are provided in Table 3. Based on the character change in the slope of the curves in Figure 6, it can be concluded that continuous laser radiation reduces the phase coherence time compared to the pulsed mode, in which the pulsed light first creates the polarization of the spin states, and the measurements themselves are carried out without exposure to light. Note that the phase coherence times under the influence of light change little with a change in temperature, while for a signal without exposure to light, the time increases by 38.4% (Table 3).

In contrast to the diverse set of spin centers in SiC, diamond contains predominantly a negatively charged *NV* center with axial and basal symmetries. A diamond crystal lattice containing only ^13^C magnetic isotopes (*I* = ½, abundance ≈ 1%) provides a stable and low-noise environment with weak spin-phonon coupling contributing to longer coherence times even at room temperature [32,33]. EPR spectrum detection when the laser is turned off, i.e., full thermodynamic equilibrium, is achieved, allowing one to obtain the proper time of transverse relaxation (*T*_2_ = 1.91 ms) for the *NV* center in the diamond (Figure 7a). In the pulsed detection mode, a similar *T*_2_ value is achieved, while continuous laser irradiation leads to a decrease in the transverse relaxation time *(T*_2_ = 1.66 ms). An interesting observation is the presence of an initial bend in the relaxation curve in the dark detection mode, which can be attributed to the spin diffusion phenomenon (Figure 7a, black squares). In the absence of optical excitation, only a part of the spin packet from the entire ensemble falls into microwave resonance, thereby creating a condition for the transfer (diffusion) of spin magnetization within the sample [34,35]. Weak spin diffusion occurs because the *NV* centers interact with each other, leading to a spin coherence time reduction. Thus, the optical excitation (CW or pulse mode) involving most of the *NV* centers introduces additional noise and decoherence mechanisms that suppress spin diffusion effects (the green and blue squares in Figure 7a).

As an additional example of the optical radiation’s influence not only on the dynamic characteristics of the spin center but also on electron–nuclear interaction features, the results of the ENDOR spectroscopy are shown (Figure 7b). Electron–nuclear interactions play a significant role in considering color centers as quantum registers since it becomes possible to create multi-level spin systems and implement long-lived quantum memory [36,37,38]. ENDOR spectra were obtained using two different detection modes (both 3072 scans): continuous laser irradiation and using a pulse sequence programmer. In the ENDOR spectrum (Figure 7b, green line), pairs of splittings caused by quadrupole interactions (labeled *QI*) from nitrogen atoms (^14^N, *I* = 1) were observed. The ENDOR spectrum with pulsed detection contains resonant RF absorption lines at 15.15 MHz, which provides valuable information on hyperfine (HFI) and quadrupole interactions [39]. However, in the case of continuous irradiation, where it seemed that the signal intensity should increase due to the higher laser power, on the contrary, within the noise limits, there is no ENDOR signal (Figure 7b, blue line). Continuous optical radiation disrupts the electron–nuclear bond of a boron vacancy in an hBN crystal by introducing excess energy that excites electrons to higher energy states, leading to a transient alteration in the local electronic environment. This excitation results in increased electron–nuclear coupling fluctuations, which, in turn, degrade the coherence of the electron spins associated with the boron vacancy.

## 4. Conclusions

This paper shows the successful development of a methodology for studying optically active spin defects (centers) within various solid-state matrices. The assembled electronic circuit, based on a 32-bit STM32F373 microcontroller (ST Microelectronics, Geneva, Switzerland), facilitates experiments in a pulsed mode with continuous semiconductor and solid-state lasers that operate across the visible to near-infrared ranges (405–1064 nm). This cost-effective device can be integrated with EPR/ENDOR spectrometers (Bruker, Karlsruhe, Germany), enabling comprehensive spin–optical and relaxation analyses of optically active spin centers. It permits the examination of the dynamic properties of photoinduced spin centers in the absence of direct exposure to optical radiation. The generated synchronized short optical pulses are sufficient to initialize the ground state of spin defects in semiconducting materials, such as SiC, diamond, and hBN, while avoiding local heating and the creation of additional defects in the crystal lattice. This advancement opens new possibilities for multi-pulse quantum operations involving optical, microwave, and radio frequency sources.

## Figures and Tables

**Figure 1 micromachines-16-00396-f001:**
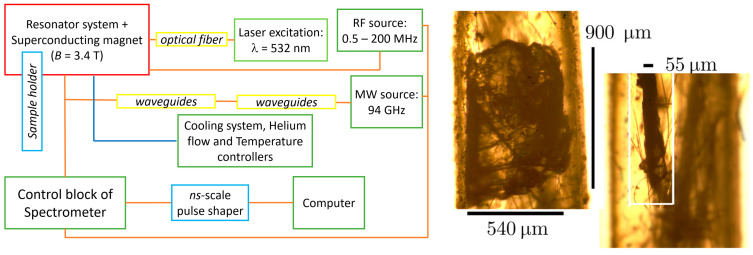
(**Left**) Measurement setup diagram including the main blocks of the spectrometer for the photoinduced EPR and ENDOR. (**Right**) hBN samples under study prepared for the high-frequency part of the spectrometer. The characteristic dimensions of the samples and capillaries correspond to the internal diameter of the resonator to achieve the highest filling factor. The white box highlights the ampoule area containing the hBN sample.

**Figure 2 micromachines-16-00396-f002:**
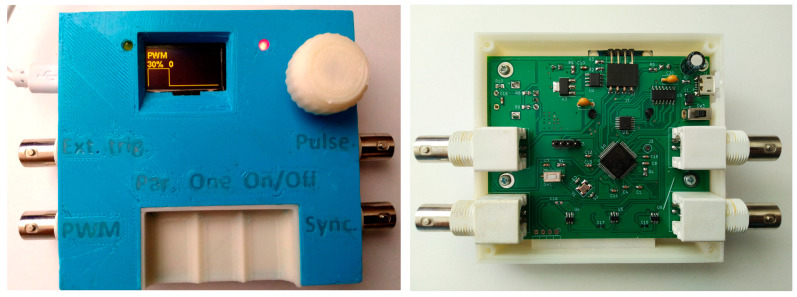
(**Left**) General view of the pulse sequence programmer. Pressing and rotating the encoder simultaneously allows for the modification of the step size for the current parameter adjustment. The scheme employs an encoder to facilitate the manual entry of parameters, utilizing a touch button designated as “Parameters” to switch into the parameter input mode, complemented by a 0.96-inch OLED screen. (**Right**) Programmer board along with the component parts of the microcontroller. Parameters can be configured through a USB interface that implements a virtual COM port.

**Figure 3 micromachines-16-00396-f003:**
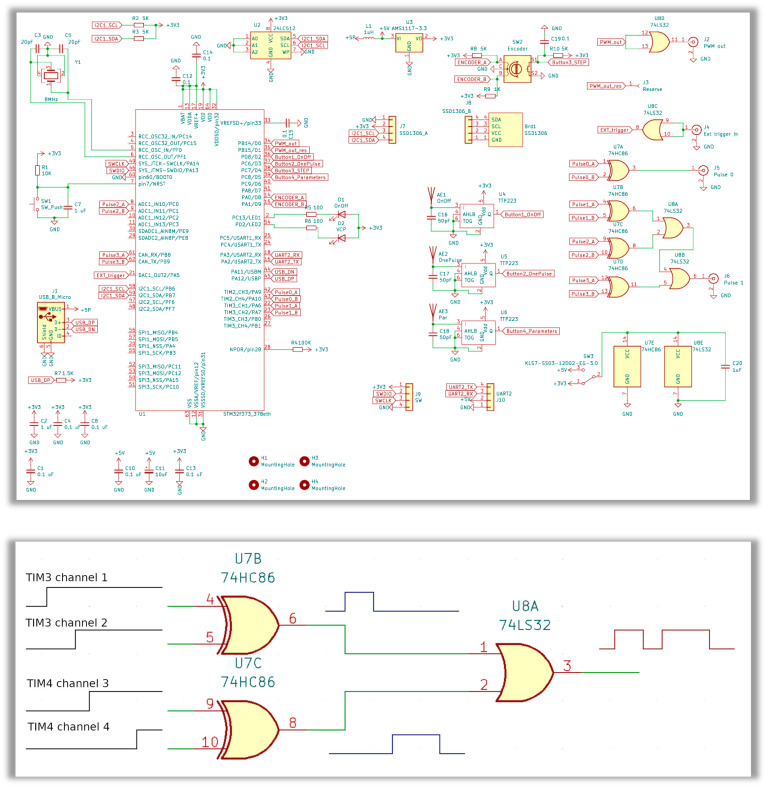
(**Upper Panel**) Complete electric schematic (circuit) diagram of a pulse sequence programmer used to perform multi-pulse experiments (for higher resolution, see Appendix A). The basis of the scheme is the U1 microcircuit of the used microcontroller with the ability to control both the U4–U6 touch buttons and the SW2 encoder, as well as via the virtual com port or the J1 connector (USB-micro type). The pulse sequence parameters are saved in the U2 EEPROM part and displayed on the OLED screen linked via the J8 connector. The U7, U8 microcircuits are used to generate TTL pulses. The SWIO J9 microcontroller programming connector is also present on the diagram. (**Bottom Panel**) Pulse generation diagram based on timer comparator signals. Lines show changes in the logic level at the microcircuit outputs over time. Four microcontroller timers are employed to generate stable pulses, independent of the load on the microcontroller CPU. The first three timers are 32-bit, while the fourth is 16-bit. The initial timer initiates the operation of the subsequent timers: the second timer produces the first pulse, the third timer generates the second pulse, and the fourth timer creates the third pulse. To facilitate the generation of three pulses, along with a synchronizing pulse, a microcircuit with four x-OR gates is utilized.

**Figure 4 micromachines-16-00396-f004:**
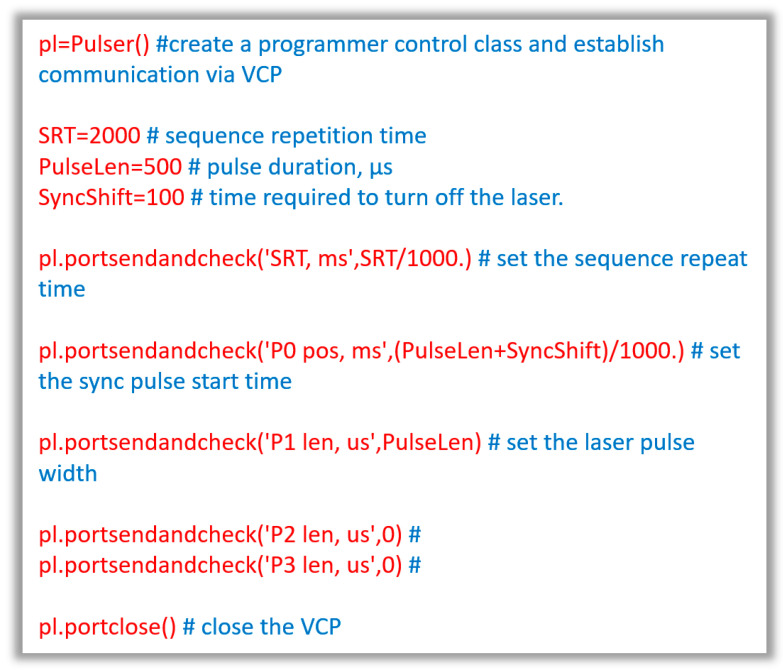
The main part of the program code in the Python language (Python 3.10 software), used for algorithmic control of the entire system (red—program text and blue—explanatory comments). The ability to control the programmer is demonstrated, and we establish communication via VCP, with parametric adjustment of all pulse sequences used.

**Figure 5 micromachines-16-00396-f005:**
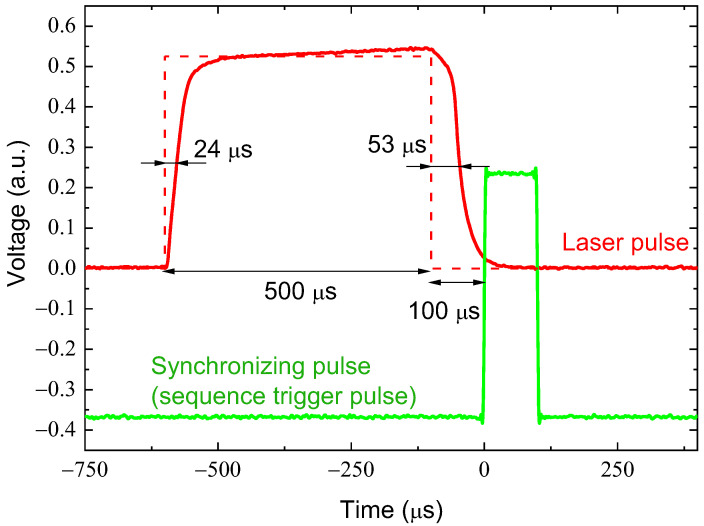
Oscillogram of laser radiation pulses (λ = 980 nm) measured by the BPW20RF diode current. Pulse parameters: 1 pulse position: 0 μs (red), duration: 500 μs; sync pulse (green) channel—position: 600 μs, duration: 100 μs. Sequence repetition time is 2 ms. The dashed line outlines the shape of the TTL driving pulse.

**Figure 6 micromachines-16-00396-f006:**
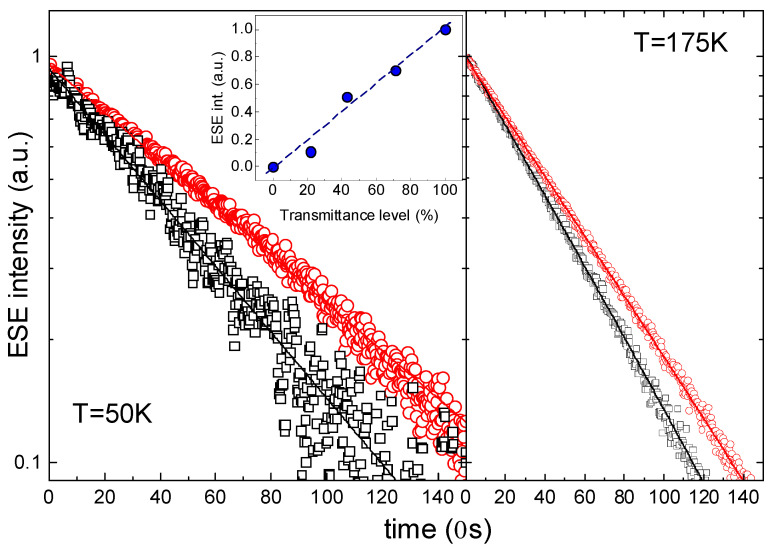
Transverse magnetization decay curves for *NV* centers in 6H-SiC crystal. Black squares—continuous laser operation. Red circles—after the laser pulse measurements. The phase coherence times of the spin packet are provided in Table 3. The insert shows the dependence of the normalized value of the integral intensity of the EPR absorption signal (blue dots) on the level of light transmission by a particular filter.

**Figure 7 micromachines-16-00396-f007:**
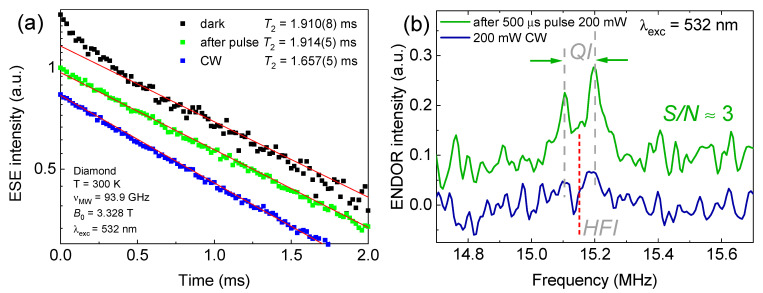
(**a**) Spin transverse magnetization decay curves of *NV* centers in diamond at *T* = 297 K. Blue squares—CW laser radiation. Green squares—pulse mode. The black squares show the ESE signal in fully “dark” mode conditions. The red straight lines emphasize the linear experimental dependence. (**b**) High-frequency (94 GHz) electron–nuclear double resonance spectrum of a boron vacancy $V_{B}^{-}$, in an hBN crystal for ^14^N nuclei of the second coordination sphere of nitrogen atoms. The results were obtained using a Mims pulse sequence at *T* = 25 K.

**Table 1 micromachines-16-00396-t001:** List of commands and programmer parameters.

Description	Keywords	Increment	Min Value	Max Value
PWM modulation frequency (kHz)	“PWM Freq, kHz”	0.1 Hz	0.1 Hz	48 kHz
Power level (%)	“Power, %”	0.1%	0%	100%
Sequence repetition time (ms)	“SRT, ms”	0.1%	0.02	2.3 × 10^9^
The position of the sync pulse in the sequence (ms)	“P0 pos, ms”	0.1%	0	“SRT, ms”
Sync pulse duration (µs)	“P0 len, μs”	0.1%	0	“SRT, ms”–“P0 pos, ms”
Position of the first pulse in the sequence (ms)	“P1 pos, μs”	0.1%	0	2.3 × 10^9^
Duration of the first pulse (µs)	“P1 len, μs”	0.1%	0	2.3 × 10^9^
Position of the second pulse in the sequence (ms)	“P2 pos, μs”	0.1%	0	2.3 × 10^9^
Duration of the second pulse (µs)	“P2 len, μs”	0.1%	0	2.3 × 10^9^
Position of the third pulse in the sequence (ms)	“P3 pos, μs”	0.1%	0	0.3 × 10^6^
Duration of the third pulse (ms)	“P3 len, μs”	0.1%	0	0.3 × 10^6^
Enable/disable synchronization on the “ExtTrig” input	“ExtTrig”	-	-	-
Maximum duty cycle for the “Pulse” output	“Max Duty”	-	0	100

**Table 2 micromachines-16-00396-t002:** Laser pulse parameters.

Laser	Turn-On Delay Time, µs	Rise Time, µs	Decay Time, µs
CNI (CNI Laser, Changchun, China), 405 nm	8 ± 0.1	4.5 ± 0.1	1.6 ± 0.1
CNI (CNI Laser, Changchun, China), 532 nm	10 ± 0.1	176 ± 0.5	41 ± 0.3
CNI (CNI Laser, Changchun, China), 980 nm	24 ± 0.1	53 ± 0.1	23.8 ± 0.1
OXlasers (OXLasers Co., Ltd., Shanghai, China), 520 nm (driver on XL1583)	2	2 ± 0.1	2 ± 0.1
LPM-1845IR-TTL, 808 nm, driver with APM9410 (CNI Laser, Changchun, China)	200 ± 0.1	68 ± 0.1	12 ± 0.1
1064T2W, 1064 nm, 2 W (CNI Laser, Changchun, China)	9000 ± 10	3600 ± 100	100 ± 10

**Table 3 micromachines-16-00396-t003:** Characteristic electron relaxation times of the *NV* centers in 6H-SiC crystal under the influence of continuous (CW) or pulsed laser radiation at temperatures of 50 K and 175 K.

Measured Value	50 K		175 K	
	CW Laser Mode	Pulse Laser Mode	CW Laser Mode	Pulse Laser Mode
Phase coherence time (*T*_2_)	53.4 ± 0.2 μs	73.9 ± 0.4 μs	50.5 ± 0.1 μs	60.1 ± 0.1 μs

## Data Availability

The original contributions presented in this study are included in the article. Further inquiries can be directed to the corresponding author.

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
