# Peer review of "Laser Pulses for Studying Photoactive Spin Centers with EPR"

_micromachines, 2025, doi:10.3390/mi16040396_

Round 1

Reviewer 1 Report

Comments and Suggestions for Authors

G. Mamin et al. present a pulse sequence programmer to be used for synchronization of laser pulses in the context of EPR experiments in photoactive centers.

The motivation is clear and solidly grounded on the work of the team in this important physical problem. The demonstration results of the new device are also very promising and highlight the importance of this technical development.

The manuscript should however be improved with respect to the description of the main object of this work: the pulse sequence programmer itself. I bring some related comments kindly asking the authors to take them into account for a revised version of the manuscript:

1) In lines 71-72, the authors indicate that “Synchronization with laser pulses was performed by feeding a synchronization signal from the programmer to the "Trig in" input of the “Pattern Jet” pulse shaper.”

I find this description unadequate and unhelpful.

2) Section 3.1 is written in a very terse and confusing way and should be clarified:

2a) The captions of Fig.2 and Fig. 3 are too much concise and little informative.

2b) Figure 3 should be mentioned much earlier than in line 123, since the ongoing description already requires it (at least from line 106 onward).

2c) In line 104 the authors mention a fourth element 2 OR, which however does not seem to be included in Fig. 2.

2d) Similarly to my point 1 above, I find the description in lines 106-108 unadequate and unhelpful for the reader.

2e) The step change column in Table 1 is written in a confusing way and there is no clarification on the text either. I believe the authors probably intend to indicate the minimum value of the step change in this column. This requires clarification, both in the title of the column and by introducing some explanation in the text as well.

3) Section 3.2 and Figure 4 also should be improved:

3a) The placement of Table 2 and Fig. 4 seems to be inverted.

3b) Fig. 4 looks like an unedited print screen from the oscilloscope and requires some work in order to be clear for the reader. The vertical and horizontal scales are not clear, as well as the identification of the blue and red line both in the figure and in the caption. The meaning and need for the numbers in the top right corner and in the bottom is not clear. The need for showing three pulses is also not clear and is not explained in the text: from the discussion in lines 146-151 it looks like it would be more clear to limit Fig. 4 to laser pulse 1 (in blue) followed by the red sync pulse.

4) Table 3 is only mentioned in the caption of Fig. 5 and is not mentioned (much less discussed) in the text.

5) The conclusion in pages 198-200 (“at low temperatures, the effect of light leads to an additional loss of quantum coherence due to the constant influx of states from the upper optical levels, the value of which is comparable to the spin-spin interaction itself”) requires additional discussion, also with respect to the known results in the literature.

Author Response

Thank you for giving me the opportunity to submit a revised draft of my manuscript titled Pulse sequence programmer for studying photoactive spin centers in various matrices to MDPI Micromashines journal. We appreciate the time and effort that you have dedicated to providing your valuable feedback on my manuscript. We are grateful to you for insightful comments on my paper. We have been able to incorporate changes to reflect most of the suggestions provided by the all reviewers. We have highlighted the changes (colored in yellow) within the manuscript. Below we will try to answer all your questions step by step.

A separate letter from the Editor of the journal informed the Authors that the article was short. In order to expand the material provided, a part of the programmer control code, a general circuit diagram of the electronic board, experimental results for NV centers in diamond and boron vacancies in a two-dimensional hBN crystal were added. Also, additional information with corresponding references was added to the existing Sections of the manuscript accordingly.

Additionally, the Authors report that they will use special language services from MDPI to improve the quality of the English language.

  1. Mamin et al. present a pulse sequence programmer to be used for synchronization of laser pulses in the context of EPR experiments in photoactive centers.

The motivation is clear and solidly grounded on the work of the team in this important physical problem. The demonstration results of the new device are also very promising and highlight the importance of this technical development.

The manuscript should however be improved with respect to the description of the main object of this work: the pulse sequence programmer itself. I bring some related comments kindly asking the authors to take them into account for a revised version of the manuscript:

1) In lines 71-72, the authors indicate that “Synchronization with laser pulses was performed by feeding a synchronization signal from the programmer to the "Trig in" input of the “Pattern Jet” pulse shaper.”

I find this description unadequate and unhelpful.

Thank you for your comment.

The corresponding changes have been made to the text of the article.

2) Section 3.1 is written in a very terse and confusing way and should be clarified:

Thank you for your comment.

Authors have added additional information to expand the section.

2a) The captions of Fig.2 and Fig. 3 are too much concise and little informative.

Thank you for your comment.

Descriptions have been added to the mentioned figures to provide more information.

2b) Figure 3 should be mentioned much earlier than in line 123, since the ongoing description already requires it (at least from line 106 onward).

The position of Figure 3 in the text has been changed and is now mentioned earlier than line 106.

2c) In line 104 the authors mention a fourth element 2 OR, which however does not seem to be included in Fig. 2.

Thank you for your comment. The authors deliberately did not show this element in the figure because it is used as a buffer.

The corresponding changes have been made to the text of the article.

2d) Similarly to my point 1 above, I find the description in lines 106-108 unadequate and unhelpful for the reader.

The authors also decided to delete this part of text.

2e) The step change column in Table 1 is written in a confusing way and there is no clarification on the text either. I believe the authors probably intend to indicate the minimum value of the step change in this column. This requires clarification, both in the title of the column and by introducing some explanation in the text as well.

The wording of the text has been corrected.

3) Section 3.2 and Figure 4 also should be improved:

Thank you for your comment. The corresponding changes have been made to the text of the article.

3a) The placement of Table 2 and Fig. 4 seems to be inverted.

The figure and table are now presented in the order of mention and logical narration.

3b) Fig. 4 looks like an unedited print screen from the oscilloscope and requires some work in order to be clear for the reader. The vertical and horizontal scales are not clear, as well as the identification of the blue and red line both in the figure and in the caption. The meaning and need for the numbers in the top right corner and in the bottom is not clear. The need for showing three pulses is also not clear and is not explained in the text: from the discussion in lines 146-151 it looks like it would be more clear to limit Fig. 4 to laser pulse 1 (in blue) followed by the red sync pulse.

Thank you for your comment.

The figure has been corrected.

4) Table 3 is only mentioned in the caption of Fig. 5 and is not mentioned (much less discussed) in the text.

The updated version of the work contains additional references to Table No. 3, including during the discussion of the results.

5) The conclusion in pages 198-200 (“at low temperatures, the effect of light leads to an additional loss of quantum coherence due to the constant influx of states from the upper optical levels, the value of which is comparable to the spin-spin interaction itself”) requires additional discussion, also with respect to the known results in the literature.

The authors agree with this comment on the need to expand the discussion by providing additional literature references.

Optical excitation can have a significant effect on the spin system under study, primarily on the dynamic (relaxation) characteristics of the color center. In addition to the direct effect of laser radiation on an individual color center, the interaction of equivalent NV centers with each other during optical pumping can lead to the formation of an additional source of spin dephasing. In this work, the high concentration of spin defects leads to a fairly close mutual arrangement of color centers and, correspondingly, accelerated spin-spin interaction. Thus, a local fluctuation of the magnetic field of a neighboring equivalent center can lead to a loss of phase coherence. The situation is further complicated by the fact that in addition to NV centers, the crystal also contains divacancy centers and silicon vacancies with a high concentration, which are photoactivated with approximately the same excitation band. Thus, the loss of electron spin coherence during continuous irradiation can be caused by the interaction of the NV center with fluctuating magnetic fields of neighboring defects of various natures, which “knock down” the phase during the detection of the ESE. The influence of optical excitation on the T2 time with two pumping modes at lower crystal temperatures (50 K) is more pronounced, since at 175 K divacancies are not observed and spin diffusion to nonequivalent centers is absent.

The wording of the article`s text has been corrected.

Reviewer 2 Report

Comments and Suggestions for Authors

The manuscript under consideration is called “Programming pulse sequences device in semiconductor lasers for studying photoactive spin centers”.
This title itself requires correction, since

- the mentioned device is not part of the lasers;

- Not all the lasers used, judging by their characteristics given in Table 2 (wavelength and rise speed), are semiconductor ones.

However, it seems to me that the content of the manuscript does not correspond to the title. The text of the manuscript only briefly mentions the “programming pulse sequences device”:  it indicates that it has a screen, buttons and a red LED, and provides a system of its commands with an indication of the ranges of values. This is all the information about the “Programming pulse sequences device”;  but instead a circuit diagram of a logical adder implemented on three discrete exclusive “OR” is given.

I do not think that the above information is a description of the device sufficient for publication.

Moreover, nowhere in the text of the manuscript is it said what exactly is the novelty of the proposed device. And what could possibly be new in a generator of a sequence of three logical pulses with given parameters?

The second part of the work, describing an experiment on measuring the relaxation time in NV centers in SiC, seems more interesting to me, although the authors eventually come to a completely trivial conclusion that “continuous laser radiation reduces the phase coherence time compared to the pulsed mode” (L193). Unfortunately, if the authors focus on this part of the article, it will turn out to be unsuitable for the “Micromashines” journal.

I also have a number of less significant comments.

  1. L60 “semiconductor solid-state lasers” - semiconductor-based lasers are generally considered as a separate class from solid-state lasers, called laser diodes, or diode lasers.
  2. The photographs in Figure 1 “Configuration of the main experimental setup together with its constituent components” are completely uninformative. Instead, a block diagram of the experimental setup with an indication of its components should be given.
  3. It is also necessary to provide a block diagram of the proposed device, possibly with a description of the algorithm of its operation.
  4. L85 “highly stable diode lasers” – as was said above, most likely not all of them are diode lasers. It is more likely that these are diode-pumped solid-state lasers. This is indicated by the characteristic wavelengths (532 and 1064 nm) and turn-on times exceeding 100 microseconds.
  5. The table headings contain many grammatically incorrect phrases, such as “The step change isn’t worse” (Table 1) or “Name laser” (Table 2).
  6. L157 “integrated intensity of the EPR signal (normalized value) strongly depends on the level of optical radiation transmission”. What “optical radiation transmission” is meant here? Perhaps the authors meant the radiation intensity?
  7. L165 “In the case of the CW laser mode, it is necessary to establish thermodynamic equilibrium, while the heating of the thermocouple junction placed in place of the sample was 40 K at 500 mW” – please clarify what was meant.
  8. L192 “it can be concluded that continuous laser radiation reduces the phase coherence time compared to the pulsed mode” – a completely trivial conclusion. Since laser radiation causes transitions from the ground to the excited state, it inevitably shortens the relaxation times.
  9. L197 “Thus, it can be assumed that at low temperatures, the effect of light leads to an additional loss of quantum coherence due to the constant influx of states from the upper optical levels, the value of which is comparable to the spin-spin interaction itself” – this conclusion does not seem correct. Rather, laser-induced transitions at both temperatures limit the transverse relaxation time to the same level (about 50 microseconds).

Overall, I believe that the article potentially contains information of interest to the reader, but it can only be published after very serious revision in accordance with the comments made above.

Comments on the Quality of English Language

Although the manuscript is generally written in clear language, a number of expressions require correction. For example, the table headings contain many grammatically incorrect phrases, such as “The step change isn’t worse” (Table 1) or “Name laser” (Table 2).

Author Response

Thank you for giving me the opportunity to submit a revised draft of my manuscript titled Pulse sequence programmer for studying photoactive spin centers in various matrices to MDPI Micromashines journal. We appreciate the time and effort that you have dedicated to providing your valuable feedback on my manuscript. We are grateful to you for insightful comments on my paper. We have been able to incorporate changes to reflect most of the suggestions provided by the all reviewers. We have highlighted (colored in yellow) the changes within the manuscript. Below we will try to answer all your questions step by step.

A separate letter from the Editor of the journal informed the Authors that the article was short. In order to expand the material provided, a part of the programmer control code, a general circuit diagram of the electronic board, experimental results for NV centers in diamond and boron vacancies in a two-dimensional hBN crystal were added. Also, additional information with corresponding references was added to the existing Sections of the manuscript accordingly.

Additionally, the Authors report that they will use special language services from MDPI to improve the quality of the English language.

The manuscript under consideration is called “Programming pulse sequences device in semiconductor lasers for studying photoactive spin centers”.

This title itself requires correction, since

- the mentioned device is not part of the lasers;

- Not all the lasers used, judging by their characteristics given in Table 2 (wavelength and rise speed), are semiconductor ones.

We agree with these comments, and therefore the title of the article should be corrected. Taking into account all the comments below, including the questions of the second reviewer, the Authors decided to focus the work on the development of a methodology (approach) for studying photoinduced centers. Thus, the revised version of the publication will have the title - Pulse sequence programmer for studying photoactive spin centers in various matrices.

However, it seems to me that the content of the manuscript does not correspond to the title. The text of the manuscript only briefly mentions the “programming pulse sequences device”:  it indicates that it has a screen, buttons and a red LED, and provides a system of its commands with an indication of the ranges of values. This is all the information about the “Programming pulse sequences device”; but instead a circuit diagram of a logical adder implemented on three discrete exclusive “OR” is given.

I do not think that the above information is a description of the device sufficient for publication.

Moreover, nowhere in the text of the manuscript is it said what exactly is the novelty of the proposed device. And what could possibly be new in a generator of a sequence of three logical pulses with given parameters?

Thank you for your comment.

We agree with this remark that the authors of the work have not fully reflected or emphasized in the text a certain novelty in the conducted study. The developed device is simple in its presentation, cheap and easy for self-assembly, requiring no additional costs. At the same time, this device can be easily integrated with an already expensive EPR spectrometer, avoiding the purchase of expensive pulsed lasers. The idea of developing an additional attachment arose when it was discovered that some photo-induced spin centers in 6H-SiC crystals, even at sufficient concentration, cannot be detected in the absence of laser excitation. At the same time, for similar systems, for example, NV centers in diamond and even in SiC of the 4H polytype, "dark" signals are observed, which at low temperatures allows determining the sign of the fine structure D (zero-field splitting). When observing "dark" spectra, it is also possible to study the dynamic (relaxation) characteristics of color centers in the absence of optical excitation, which provides additional information on the mechanisms of coherent dephasing of the defect. In the case of spin centers in a 6H-SiC crystal, such opportunities for obtaining valuable information are lost, since the signal without optical excitation is completely absent, regardless of the crystal temperature or detection conditions (CW or ESE). In this case, the novelty of the work is concentrated in the development of a methodology for studying such spin centers with maximum polarization.

The developed device will allow experiments to be carried out:

- With extremely insignificant local heating of the studied crystal in the resonator;

- Synchronization of the optical pulse with the sequence for detecting the electron spin echo is ensured;

- A modulation system (PWM) is present as an additional option; it is useful if synchronization is not required;

- Low-power pulse excitation allows avoiding the creation of new defect centers, as opposed to femtosecond lasers;

- Also possible for use in NMR spectroscopy as an extension of experimental capabilities.

The second part of the work, describing an experiment on measuring the relaxation time in NV centers in SiC, seems more interesting to me, although the authors eventually come to a completely trivial conclusion that “continuous laser radiation reduces the phase coherence time compared to the pulsed mode” (L193). Unfortunately, if the authors focus on this part of the article, it will turn out to be unsuitable for the “Micromashines” journal.

We fully agree with this remark.

The authors aimed to show the experimental results exclusively as an application of how the pulse sequence programmer can be used to obtain information about the spin system of color centers. The effect of optical radiation on the T2 time was given in the article as an example. Despite the apparent simplicity of the measurements, the EPR signal for NV centers in the 6H-SiC crystal is completely absent when the laser is turned off. Thus, valuable information about the dynamic characteristics of spin defects in the absence of external optical action is lost. However, using the developed cheap programmer, the Authors managed to register a "dark" signal. Additional experiments were also carried out for the spin-lattice relaxation time T1 and establishing the degree of polarization depending on the exciting optical pulse. However, the Authors similarly assume that by focusing on the experimental results, the content of the publication will be all the fields of interest of the journal "Micromashines". However, to expand this concept, results on NV centers in nanodiamonds and highly promising boron vacancy centers in a hexagonal boron nitride crystal (a two-dimensional material) were additionally included. For the case of a boron vacancy in hBN, data are shown for the high-frequency electron-nuclear double resonance method, which allows one to obtain information on hyperfine and quadrupole interactions.

I also have a number of less significant comments.

L60 “semiconductor solid-state lasers” - semiconductor-based lasers are generally considered as a separate class from solid-state lasers, called laser diodes, or diode lasers.

Corrected.

The photographs in Figure 1 “Configuration of the main experimental setup together with its constituent components” are completely uninformative. Instead, a block diagram of the experimental setup with an indication of its components should be given.

It is also necessary to provide a block diagram of the proposed device, possibly with a description of the algorithm of its operation.

Additional information has been added.

L85 “highly stable diode lasers” – as was said above, most likely not all of them are diode lasers. It is more likely that these are diode-pumped solid-state lasers. This is indicated by the characteristic wavelengths (532 and 1064 nm) and turn-on times exceeding 100 microseconds.

The table headings contain many grammatically incorrect phrases, such as “The step change isn’t worse” (Table 1) or “Name laser” (Table 2).

Corrected.

L157 “integrated intensity of the EPR signal (normalized value) strongly depends on the level of optical radiation transmission”. What “optical radiation transmission” is meant here? Perhaps the authors meant the radiation intensity?

The wording of the text has been corrected.

In this experiment, suitable filters with different transmission values (levels) were used to change the intensity of laser excitation.

Revised version:

integrated intensity of the EPR signal (normalized value) strongly depends on the level of optical radiation intensity…

L165 “In the case of the CW laser mode, it is necessary to establish thermodynamic equilibrium, while the heating of the thermocouple junction placed in place of the sample was 40 K at 500 mW” – please clarify what was meant.

The wording of the text has been corrected.

In the absence of helium flow cooling, when exposed to this laser in continuous mode, the crystal temperature is heated by 40 K.

L192 “it can be concluded that continuous laser radiation reduces the phase coherence time compared to the pulsed mode” – a completely trivial conclusion. Since laser radiation causes transitions from the ground to the excited state, it inevitably shortens the relaxation times.

The wording of the text has been corrected.

Additional experimental results are added, which serve as a demonstration of the applicability of the pulse sequence programmer to study the coherence properties of various spin centers. In the discussion, the authors also described in more detail the mechanisms that can influence spin dephasing under laser irradiation.

L197 “Thus, it can be assumed that at low temperatures, the effect of light leads to an additional loss of quantum coherence due to the constant influx of states from the upper optical levels, the value of which is comparable to the spin-spin interaction itself” – this conclusion does not seem correct. Rather, laser-induced transitions at both temperatures limit the transverse relaxation time to the same level (about 50 microseconds).

The authors agree with this comment on the need to expand the discussion by providing additional literature references.

Optical excitation can have a significant effect on the spin system under study, primarily on the dynamic (relaxation) characteristics of the color center. In addition to the direct effect of laser radiation on an individual color center, the interaction of equivalent NV centers with each other during optical pumping can lead to the formation of an additional source of spin dephasing. In this work, the high concentration of spin defects leads to a fairly close mutual arrangement of color centers and, correspondingly, accelerated spin-spin interaction. Thus, a local fluctuation of the magnetic field of a neighboring equivalent center can lead to a loss of phase coherence. The situation is further complicated by the fact that in addition to NV centers, the crystal also contains divacancy centers and silicon vacancies with a high concentration, which are photoactivated with approximately the same excitation band. Thus, the loss of electron spin coherence during continuous irradiation can be caused by the interaction of the NV center with fluctuating magnetic fields of neighboring defects of various natures, which “knock down” the phase during the detection of the ESE. The influence of optical excitation on the T2 time with two pumping modes at lower crystal temperatures (50 K) is more pronounced, since at 175 K divacancies are not observed and spin diffusion to nonequivalent centers is absent.

The wording of the text has been corrected.

Round 2

Reviewer 1 Report

Comments and Suggestions for Authors

The authors have replied in a most satisfactory way to all the queries. I support the publication of this work in "Micromachines".

Author Response

Dear Reviewer

Thank you very much again for providing valuable comments to improve the scientific quality of the article!

Reviewer 2 Report

Comments and Suggestions for Authors

I am glad to see that the Authors took the reviewer's comments into account. Moreover, the changes made by the Authors to the manuscript are so extensive that, in fact, it now represents a new article and requires a new full review.

So also have a number of comments on this new article.

  1. The structure of the manuscript has suffered. In particular, the authors added a large fragment of text to the introduction (L43-L83) describing the materials under study. Perhaps this was done at the request of other reviewers, but I find this description redundant, since it does not fully correspond to the main topic of the article, i.e., the description of the pulse programmer. This impression is reinforced when reading the new fragment (L350-L419), which, in my opinion, should be moved to the introduction, shortened and structured. In the final version, the description of the properties of the samples should not significantly exceed the description of the device to which the article is devoted.
  2. The new version of the title sounds incomprehensible, since neither the EPR method nor lasers are mentioned in the title. The error in the 1st version was not in the mention of lasers, but in their incorrect mention.
  3. L106: “The assembled electronic circuit can be integrated with semiconductor lasers across a wide range of wavelengths” – here the authors replaced word “solid-state“ with “semiconductor”, while, judging by the parameters, they use lasers of both types.
  4. L131: “To detect EPR spectra a Bruker Elexsys E680 spectrometer operating in pulse mode was used” – this is a repeat of the previous sentence.
  5. L157 “A Chilasers firm laser” sounds ambiguous.
  6. L213: ”Parameters can be input through a USB interface that implements a virtual COM port” – maybe “configured” is better than “input”?
  7. L259 “photodiode operating in current generation mode” - there are two modes of photodiode operation: photovoltaic mode and photoconductive mode. In the first mode, the photodiode generates voltage, in the second it conducts current under the influence of the applied voltage.
  8. Fig. 5 – it is not clear from the caption which pulses are meant, the picture itself is not very informative.
  9. L303 “strongly depends on the level of optical radiation transmission” – it is still not clear from the text what transmission is meant. It would be more correct to write about the intensity of the incident light.
  10. Captions to Fig. 6,7 – which sample is meant?
  11. L409: “leading to a broadening of the spin coherence time (T2)” – either “to the line broadening”, or “to a decrease in the spin coherence time”.
  12. L417: “Thus, the presence of optical excitation can significantly alter the spin dynamics of NV centers, suppressing the effects of spin diffusion observed in dark mode detection” – this does not follow from the previous text. Please clarify.

To sum up, I believe that the text needs to be further revised.

Author Response

Dear Reviewer

Thank you again very much for your valuable comments with well-founded claims and extremely important suggestions. The quality of our article has increased many times thanks to your advice and comments. Your contribution to the improvement of our scientific article is invaluable.

Reviewer's comments

I am glad to see that the Authors took the reviewer's comments into account. Moreover, the changes made by the Authors to the manuscript are so extensive that, in fact, it now represents a new article and requires a new full review.

So also have a number of comments on this new article.

The structure of the manuscript has suffered. In particular, the authors added a large fragment of text to the introduction (L43-L83) describing the materials under study. Perhaps this was done at the request of other reviewers, but I find this description redundant, since it does not fully correspond to the main topic of the article, i.e., the description of the pulse programmer. This impression is reinforced when reading the new fragment (L350-L419), which, in my opinion, should be moved to the introduction, shortened and structured. In the final version, the description of the properties of the samples should not significantly exceed the description of the device to which the article is devoted.

The author fully agrees with this remark. The text (L350-L419) was moved to the Introduction merged with (L43-L83) with the significant reduction of the content. The discussions of the results (L350-L419) were also revised to remove unnecessary details, leaving concise conclusions. Figures 7 and 8 have been combined into Figures 7a and 7b. In the updated version of the Materials and Methods section, information on synthesis and descriptions of samples has been significantly reduced. Two subsections have been added: 2.1. Methodology and 2.2. Samples.

The new version of the title sounds incomprehensible, since neither the EPR method nor lasers are mentioned in the title. The error in the 1st version was not in the mention of lasers, but in their incorrect mention.

Updated title: Laser pulses for studying photoactive spin centers with EPR

L106: “The assembled electronic circuit can be integrated with semiconductor lasers across a wide range of wavelengths” – here the authors replaced word “solid-state“ with “semiconductor”, while, judging by the parameters, they use lasers of both types.

The authors apologize for the confusing information provided. Both semiconductor and solid-state lasers were used in this work. The sentence has been corrected:

… integrated with both semiconductor and solid-state lasers across a wide …

L131: “To detect EPR spectra a Bruker Elexsys E680 spectrometer operating in pulse mode was used” – this is a repeat of the previous sentence.

Thank you very much. The error with the repeated mention has been corrected.

L157 “A Chilasers firm laser” sounds ambiguous.

The authors agree with this comment. The corresponding text has been corrected:

For the main experimental EPR section of this article, lasers with wavelengths of 980 nm (diode laser) and 532 nm (diode-pumped solid-state) were employed, both of which were manufactured by Changchun New Industries (CNI) Optoelectronics Tech. Co., Ltd. (China).

L213: ”Parameters can be input through a USB interface that implements a virtual COM port” – maybe “configured” is better than “input”?

The authors agree with this suggestion. The corresponding text has been corrected:

…Parameters can be configured through a USB interface that implements a virtual COM port

L259 “photodiode operating in current generation mode” - there are two modes of photodiode operation: photovoltaic mode and photoconductive mode. In the first mode, the photodiode generates voltage, in the second it conducts current under the influence of the applied voltage.

In this work, the photovoltaic mode (voltage generation) of operation of the photodiode is used.

Corrected text:

… photodiode operating in a photovoltaic mode (voltage generation), exhibiting …

Fig. 5 – it is not clear from the caption which pulses are meant, the picture itself is not very informative.

The figure has been 5 corrected with the addition of some explanatory information.

L303 “strongly depends on the level of optical radiation transmission” – it is still not clear from the text what transmission is meant. It would be more correct to write about the intensity of the incident light.

The corresponding part of the sentence has been corrected:

… on the level of intensity of the incident light.

Captions to Fig. 6,7 – which sample is meant?

New information regarding the samples used has been added to the Figure 6 and 7 descriptions.

L409: “leading to a broadening of the spin coherence time (T2)” – either “to the line broadening”, or “to a decrease in the spin coherence time”.

The authors agree with this comment. The sentence has been revised:

Weak spin diffusion occur because the NV centers interact with each other leading to a spin coherence time reducing.

L417: “Thus, the presence of optical excitation can significantly alter the spin dynamics of NV centers, suppressing the effects of spin diffusion observed in dark mode detection” – this does not follow from the previous text. Please clarify.

The corresponding text has been corrected:

Thus, the optical excitation (CW or pulse mode) involving most of the NV centers introduce additional noise and decoherence mechanisms that suppress spin diffusion effects (green or blue squares on Figure 7a).

To sum up, I believe that the text needs to be further revised.
